# Consistent Explanations in the Face of Model Indeterminacy via Ensembling

Dan Ley    Leonard Tang    Matthew Nazari    Hongjin Lin    Suraj Srinivas    Himabindu Lakkaraju [1]

## Abstract

This work addresses the challenge of providing consistent explanations for predictive models in the presence of model indeterminacy, which arises due to the existence of multiple (nearly) equally well-performing models for a given dataset and task. Despite their similar performance, such models often exhibit inconsistent or even contradictory explanations for their predictions, posing challenges to end users who rely on them to make critical decisions. Recognizing this, we introduce ensemble methods as an approach to enhance the consistency of the explanations provided in these scenarios. Leveraging insights from recent work on neural network loss landscapes and mode connectivity, we devise ensemble strategies to efficiently explore the *underspecification set* – the set of models with performance variations resulting solely from changes in the random seed during training. Experiments on five benchmark financial datasets reveal that ensembling can yield significant improvements when it comes to explanation similarity, and demonstrate the potential of existing ensemble methods to explore the underspecification set efficiently. Our findings highlight the importance of considering model indeterminacy when interpreting explanations and showcase the effectiveness of ensembles in enhancing the reliability of explanations in machine learning.

## 1. Introduction

The rapidly increasing adoption of machine learning (ML) models in a wide range of applications, including healthcare, finance, and criminal justice, underscores the need for transparency and trust in automated decision-making. However, ensuring the consistency of explanations offered by these models has proven to be a complex challenge with far-reaching implications, particularly in high-stakes scenarios, where decisions carry a substantial impact on individuals and society. This has led to the introduction of various regulatory principles (AI-Rights, 2022; GDPR, 2018).

A key factor complicating the interpretability of predictive models is model indeterminacy, which arises from the existence of multiple (nearly) equally well-performing models for a given dataset and task. Despite comparable performance, these models often provide inconsistent or even contradictory explanations for their decisions (Figure 1). Such *explanatory multiplicity* can severely undermine transparency efforts, erode trust in automated decision systems, or lead to potentially harmful outcomes in critical applications such as risk assessment or medical diagnosis.

For instance, in credit lending, a predictive model may be employed to determine a person's creditworthiness. In this context, model indeterminacy can lead to inconsistent explanations for different loan approvals or rejections. An individual might be denied a loan based on one model's decision, while a nearly equally performing model might provide a different explanation and approve the loan. In addition, if a model is periodically retrained without accounting for indeterminacy, it may further exacerbate inconsistencies and erode trust in the decision-making process.

This prompts us to investigate model indeterminacy in more depth. Specifically, our work examines the *underspecification set* (Brunet et al., 2022; D'Amour et al., 2022) – the group of models whose performance variations arise solely from changes to the random seed used in training. Equally performing models in this set can each offer markedly different explanations for a given input, and prior work has also indicated a lack of correlation between the prediction of a model and its explanation (Black et al., 2021; Brunet et al., 2022). This raises some important questions. *Do equally performing ML models share common patterns in their explanations, or is explanatory multiplicity largely arbitrary? Do there exist systematic approaches that can be utilized to align explanations, without compromising performance?*

Motivated by these questions, we propose the use of ensemble methods to mitigate explanatory multiplicity among equally performing neural networks (Figure 1), and explore the application of various ensemble methods towards improving the consistency of explanations generated. Focusing

---

[1]Harvard University, Cambridge, MA. Correspondence to: Dan Ley <dley@g.harvard.edu>.

*Workshop on Interpretable ML in Healthcare at International Conference on Machine Learning (ICML)*, Honolulu, Hawaii, USA. 2023.

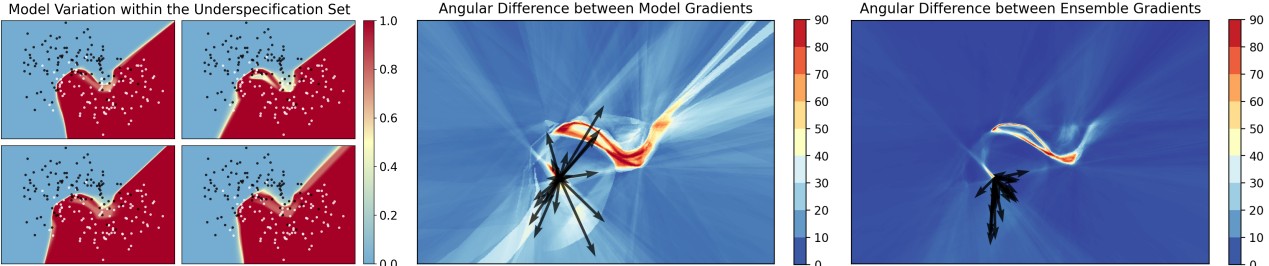

Figure 1: Illustration of explanatory multiplicity between members of the underspecification set (model variation due only to random seed). **Left:** softmax probabilities of neural networks trained on the two moons dataset, with test points depicted in black and white. All models achieve similar performance on test data. **Center:** average pairwise angular difference between model explanations in the same region of input space. Gradients with respect to the input, a proxy for many explanation techniques, are shown for a test point of high disagreement (i.e. high angular difference) between models. **Right:** average pairwise angular difference between ensemble explanations. Ensembles consist of 10 constituent models sampled from the underspecification set. Our findings indicate that *ensembling results in model alignment, promoting agreement between explanations in input space.*

on neural networks is particularly relevant to this problem setup, given their expressivity and ability to fit complex functions. Neural networks of sufficient size can cover a large range of predictors, even for relatively small tasks. Our aim is to devise efficient ensembling strategies that can align explanations across various diverse modes, while simultaneously requiring fewer pre-trained models to do so. Our methods are informed by previous research on neural network loss landscapes and existing explanation techniques. More specifically, our overall contributions are as follows:

1. In §3, we propose ensemble strategies that target two facets of the neural network loss landscape: local perturbations on model weights (§3.1); and global connections between models along paths of near-constant loss (§3.2). We provide a toy illustration to demonstrate how explanatory multiplicity can be mitigated through both types of exploration (§3.3).

2. In §4.1, we perform ablations to understand the impact of weight perturbations on explanations, finding that it depends critically on the layer being perturbed.

3. In §4.2, we conduct experiments across five benchmark financial datasets and three commonly used explanation techniques, demonstrating that ensembles constructed through local or global exploration of the loss landscape can yield significant improvements in terms of average pairwise top-$k$[1] similarity between explanations. We showcase how a novel combination of both local and global ensembling can yield performance improvements up to fivefold over naive ensembling, given a fixed number of pre-trained models.

While our experiments highlight the potential of ensembles in alleviating the challenges posed by model indeterminacy, the problem of providing consistent explanations for ML

models remains an ongoing area of research. This work seeks to contribute to the development of more trustworthy and reliable ML systems that can be implemented responsibly in high-stakes, real-world applications, and provides an empirical analysis of ensemble behavior in such regards.

## 2. Related Work

### 2.1. Underspecification and Model Indeterminacy

D'Amour et al. (2022) identified underspecification in ML pipelines as a key reason for poor deployment behavior, defining an ML pipeline to be underspecified when it can return various distinct predictors with equivalently strong test performance. They identify the resulting instability as a distinct failure mode from issues arising from structural mismatches between training and deployment domains. Similarly, Brunet et al. (2022) investigated the implications of model indeterminacy on post-hoc explanations of predictive models, demonstrating that underspecification can lead to significant explanatory multiplicity, and highlighting that predictive multiplicity and epistemic uncertainty are not reliable indicators of explanatory multiplicity. Both works motivate the need to explicitly account for underspecification in ML pipelines intended for real-world deployment.

### 2.2. Explanation Methods

Various explanation techniques have been proposed to provide interpretability for ML models, with some of the most common being Saliency (Simonyan et al., 2013), its smoothed counterpart SmoothGrad (Smilkov et al., 2017), as well as local function approximators such as LIME (Ribeiro et al., 2016) and SHAP (Lundberg and Lee, 2017). Smoothgrad has been highlighted as particularly faithful to the model being explained (Agarwal et al., 2022). Recent efforts have been lent toward understanding the workings and limitations of these methods. For instance, Krishna et al.

---

[1]The $k$ features in an explanation with largest absolute value.

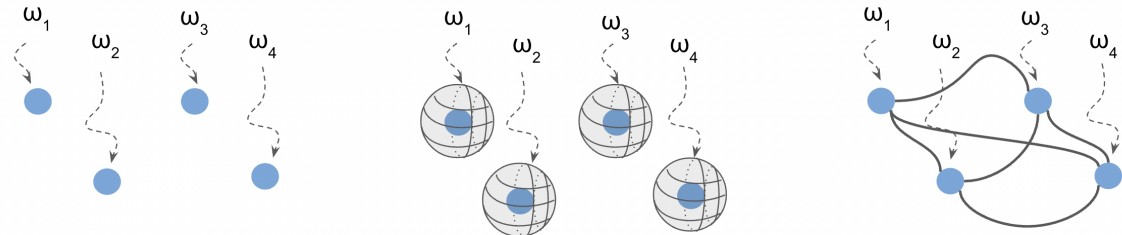

Figure 2: Weight space visualizations of the methods we explore. **Left:** the underspecification set of (nearly) equally performing models, trained with the same hyperparameters but different random initializations. **Center:** local *weight perturbations* to each of the models in the underspecification set, forming regions of alternate predictors of similar performance. The perturbations are themselves ensembled to form *perturbed models*. **Right:** global *mode connectivity* paths of near-constant loss, connecting different members of the underspecification set. Vanilla ensembling of models increases explanation consistency but requires an impractically large number of models. Our methods leverage ensembles formed by local weight perturbations and global mode-connected models to achieve better explanation consistency with fewer models and computational overhead.

(2022) analyzed the disagreement between different explanation methods for a fixed model, highlighting the challenge of providing consistent explanations, while Han et al. (2022) attempted to unify popular post-hoc explanation methods.

### 2.3. Deep Ensembles

Ensemble methods have traditionally served as potent tools within ML, providing advantages such as reduced generalization error (Allen-Zhu and Li, 2023; Dieterich, 2000), robustness to adversarial examples (Yang et al., 2021), and uncertainty estimation (Lakshminarayanan et al., 2017). However, analysis of ensembles alongside explanatory multiplicity has only been briefly touched. Black et al. (2021) introduced selective ensembles, which abstain from decisions in regions of predictive uncertainty, in an effort to avoid contradictory predictions and make explanations more consistent, though with limited analysis on explanations explicitly. Li et al. (2021) also proposed a cross-model consensus of explanations to identify common features used by various models for classification, finding correlations between consensus score and model performance for vision tasks.

### 2.4. Loss Landscapes and Mode Connectivity

We are interested in leveraging known properties of neural networks to combat model indeterminacy and expedite ensembling of the underspecification set. A key aspect in this regard is understanding where high performing models are located within the loss landscape. These were traditionally viewed as lying in disjoint local minima, until Garipov et al. (2018) and Draxler et al. (2019) demonstrated that these minima could be connected via paths of near constant loss in weight space, a concept dubbed *mode connectivity*. This surprising result opened the door for a body of subsequent research, including applications of mode connectivity in adversarial robustness (Zhao et al., 2020), discovery of mode connecting volumes (Fort and Jastrzebski, 2019; Ben-

ton et al., 2021), and alignment of models in weight space through permutation symmetries (Ainsworth et al., 2023; Singh and Jaggi, 2020; Tatro et al., 2020). Although ensemble methods feature prominently in this literature, their application with respect to explanations has received very little attention. Model indeterminacy and advances in loss landscape understanding have emerged as recent, yet distinct developments, with the two existing almost in parallel. The aim of this work is to serve as a catalyst for their convergence, and bring about a unified exploration of both areas.

## 3. Ensemble Strategies

Inconsistency between explanations of different ML models poses a significant challenge, where choices as arbitrary as the random seed used during the training phase of a deployed ML model can drastically affect the explanation provided to a particular individual (Brunet et al., 2022; D'Amour et al., 2022). Ensembling provides a promising avenue for aligning explanations across similarly performing models. However, conventional or *vanilla* methods may necessitate a prohibitively large number of models to do so. In this context, the primary question driving our research is:

*Can we maximize the explanation similarity between two ensembles constructed from $n$ samples of the underspecification set?*

This question underscores our exploration of ensemble methods towards bridging the gap between computational efficiency and explanation consistency. Once a sufficiently large underspecification set has been formed for a given dataset and task, samples of equally performing models are used to construct ensembles. We proceed to outline our ensembling strategies (summarized in Figure 2).

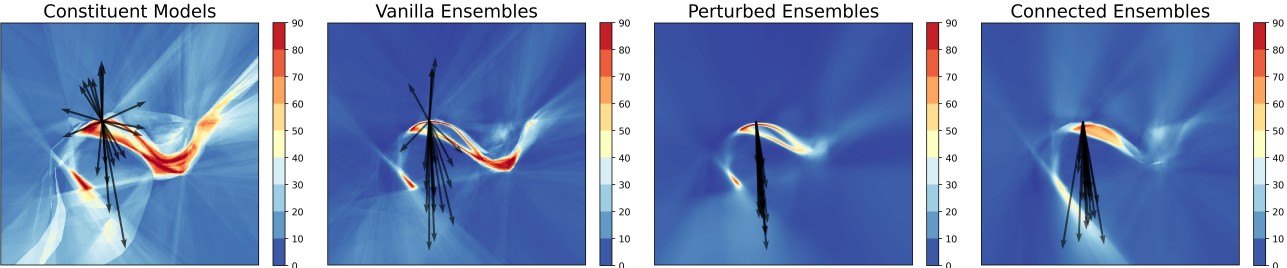

Figure 3: Heatmaps of explanation disagreement (average pairwise angular difference between model gradients) for the described ensembling techniques, each constructed from four pre-trained models. **Left:** Baseline disagreements between equally performing models. Observe high disagreement between gradients (dark arrows) for a given test point. **Left Center to Right:** the efficacy of vanilla ensembles, ensembles via weight perturbation, and ensembles via mode connectivity on aligning explanations. For the same number of pre-trained models, local perturbations or global connections demonstrate potential for superior alignment of gradients between ensembles.

### 3.1. Ensembles via Weight Perturbation

While we find that standard ensembling is capable of aligning the explanations of models from the underspecification set, this often requires a large collection of pre-trained models. To reduce the number of models needed, we propose a computationally cheap method that injects Gaussian noise $\epsilon \sim \mathcal{N}(0, \sigma^2)$ to the weights or biases of a given layer. Figure 2, Center, depicts this in relation to the other methods.

Constructing an ensemble using this method requires the weights of $n$ pre-trained models sampled from the underspecification set, each denoted $\omega_i$. The method perturbs each $\omega_i$ a total of $m$ times, leading to $m$ variants of each original model $f_{\omega_i}$. We denote the collection of the $m$ variants as the *perturbed model* $f_{\tilde{\omega}_i}$. Generating $n$ of these perturbed models does not involve the computational expense of training $n \times m$ distinct models from scratch. Instead, we capitalize on the learned knowledge of the $n$ pre-trained models, exploring the local regions around their weights.

We draw inspiration from Smoothgrad (Smilkov et al., 2017), which perturbs the input to stabilize gradient estimation, providing an explanation of a smoothed approximation to a noisy decision boundary. Our approach, in contrast, smooths both the explanation and the boundary. Techniques that perturb model parameters have been explored before, in adversarial training (Wu et al., 2020), robust explanation generation (Upadhyay et al., 2021), and more generally in Bayesian learning (Gal, 2016; MacKay, 1992). Other conceptually similar examples are dropout (Srivastava et al., 2014), and stochastic weight averaging (Izmailov et al., 2018). Informed by these works, we introduce weight perturbation as a computationally cheap ensembling strategy, with the specific goal of enhancing explanation similarity.

### 3.2. Ensembles via Mode Connectivity

Instead of sampling locally around a single model in weight space, we also consider global explorations between pairs of models along constant paths of low loss found via mode con-

nectivity (Garipov et al., 2018). Quadratic Bezier curves provide a convenient parametrization of smooth paths, where the curve denoted $\phi_\theta(t)$ with endpoints $\omega_1$ and $\omega_2$ (sampled from the underspecification set) is given by

$$\phi_\theta(t) = (1-t)^2\omega_1 + 2t(1-t)\theta + t^2\omega_2, 0 \le t \le 1$$

Once such paths between pairs of models are obtained, ensembles can be constructed at little extra cost, most simply by uniformly sampling the scalar $t$, returning model weights along the path (Figure 2, Right). To construct an ensemble from $n$ pre-trained models, we form connections between $n/2$ non-overlapping pairs, before sampling along these connections and ensembling the sampled weights. We discuss the curve finding procedure in §4.

### 3.3. Illustration on a Toy Example

In Figure 3, we illustrate the problem of explanatory multiplicity on the two moons dataset, and visualize the effectiveness of the described ensembling techniques in aligning the input gradients (saliency) of models. Observe that while vanilla ensembles (Left Center) constructed from four pre-trained models $f_{\omega_i}$ demonstrate reductions in average pairwise explanation difference compared to single models (Left), ensembles constructed from four perturbed models $f_{\tilde{\omega}_i}$ effectively smooth the explanation distribution, achieving significantly better alignment (Right Center). We note similar smoothing effects for ensembles constructed via mode connectivity (Right), again using four pre-trained models i.e. two mode connected pairs.

The process of averaging the outputs of local weight perturbations can be viewed as a mapping from the underspecification set of original models, to a reduced set of perturbed models, characterized by lower gradient variance. This *shrinking* process necessitates fewer perturbed models to reach convergence in explanation alignment. In contrast, mode connectivity provides a more extensive traversal of the underspecification set, charting a broad range along the path

between two models. The following experiments demonstrate the efficacy of both exploratory methods in real-world scenarios. We find that their relative performance often hinges upon the unique properties of the given dataset, the explanation technique, and the particular model class used.

## 4. Experiments

To evaluate the effectiveness of these strategies in maximizing explanation similarity between ensembles, we conduct experiments spanning five benchmark financial datasets, three similarity metrics (each with a focus on distinct aspects of explanation similarity), and three common explanation methods. In this section, we outline the technical details involved. In §4.1, we perform ablations to understand the impact of local weight perturbations. In §4.2 we, showcase the merits of both methods in maximizing the expected explanation similarity between any two ensembles constructed from $n$ samples of the underspecification set, improvements attained without compromising test accuracy. Our experiments were conducted in Python (Van Rossum and Drake Jr, 1995) on standard consumer-grade CPUs,[2] parallelized with `joblib` (Joblib Development Team, 2020).

**Datasets**   We use five publicly available financial datasets: the Home Equity Line of Credit (HELOC) (FICO, 2018) and German Credit (Dua and Graff, 2017) datasets both measure forms of credit risk, given input information regarding a loanee; Adult Income (Dua and Graff, 2017) is used to predict whether a person's salary exceeds a threshold of $50,000; Default Credit (Yeh and Lien, 2009) and Give Me Some Credit (GMSC) (Freshcorn, 2022) involve predicting the likelihood of a borrower defaulting on a loan, based on a variety of financial and personal features. Categorical features are one-hot encoded, and data is normalized to zero mean and unit variance, in part to ensure that top-$k$ feature explanations are meaningful, and not skewed towards features with smaller ranges. We fix a split of 80% of each dataset for training, and 20% to evaluate explanations and test accuracy. Full details are provided in Appendix A.1.

**Model selection**   Our experiments use `PyTorch` (Paszke et al., 2019) to implement feedforward neural networks with fixed architectures: hidden layers of size 128, 64, and 16 are used, to provide sufficient capacity for the networks to capture largely diverse, but equally performing functional solutions. We use ReLU activations for the intermediate layers and Softmax for the output. Model selection sweeps are performed with `RayTune` (Liaw et al., 2018), to identify the best performing model hyperparameters for a given dataset. Full details are provided in Appendix A.2.

---

[2]Specifically, an 8-core/8-thread Apple M1 Pro chip and a 12-core/24-thread AMD Ryzen 9 5900X processor.

To explore the underspecification set, we train 1000 models using the optimal hyperparameters found, changing only the random seed that controls the initialization of each neural network in weight space. We decide against randomly shuffling the training data between epochs, focusing on the effect of underspecification solely in terms of the random initialization. We do not employ practices such as weight decay or dropout, as these techniques could limit the expressivity of the models and consequently constrain the variability of explanations, obscuring our ability to test the effectiveness of ensembling strategies in aligning diverse explanations.

**Explanations**   We experiment with three common explanation methods. Input gradients, or Saliency (Simonyan et al., 2013), Smoothgrad (Smilkov et al., 2017), and DeepSHAP (Lundberg and Lee, 2017). Specifically, we seek to verify if our method offers benefits beyond those provided by Smoothgrad, given its conceptual similarity to weight perturbation. Given the widespread use of SHAP, we experiment on DeepSHAP, which uses a selection of background samples to approximate the conditional expectations of SHAP, described in Lundberg and Lee (2017). In the interests of compute, we limit the number of test inputs being explained with DeepSHAP to 100. We note that the same train/test samples are used for a given dataset, and that DeepSHAP values reached convergence, eliminating any potential source of variation in explanations themselves.

**Explanation similarity metrics**   We are interested in comparing the explanations provided by two models for a given input, and adopt existing top-$k$ metrics to cover distinct forms of agreement. Each metric takes a pair of top-$k$ feature sets corresponding to the two explanations.

*Sign-Agreement (SA)* computes the fraction of top-$k$ features that appear in both explanations and share the same sign (Krishna et al., 2022), and can be viewed as the overlap between two top-$k$ explanations, after accounting for the direction of features (whether they contributed positively or negatively to the model's prediction). We modify two other metrics taken from (Brunet et al., 2022):

*Signed-Set Agreement (SSA)* is binary (per-sample), and is 1 when the two model's top-$k$ feature sets contain the same features and the features have the same signed value (the order of the top-$k$ features does not matter). SSA is a stricter form of SA, and is satisfied if and only if SA evaluates to 1.

*Consistent Direction of Contribution (CDC)* is also binary (per-sample) and requires that any feature that appears in the top-$k$ of one model has the same signed value if it appears in the other model. CDC estimates the likelihood of a top-$k$ explanatory feature changing its contribution direction when switching between pairs of roughly equivalent models. Such changes can be confusing for end users or practitioners.

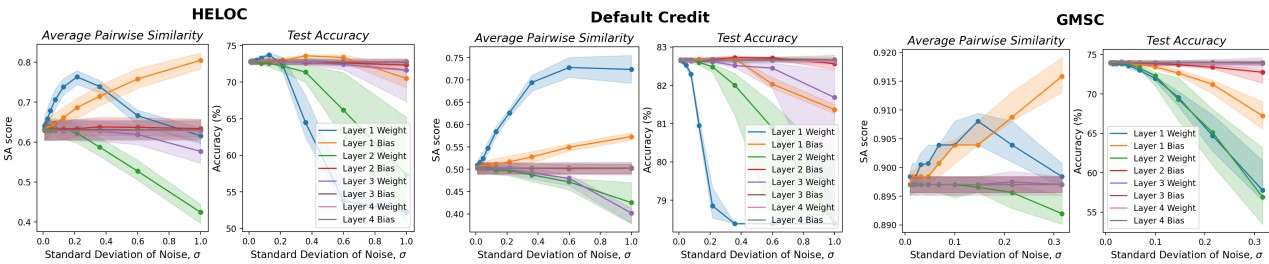

Figure 4: Effect of $\sigma$ on top-k SA scores (for input gradients and $k = 5$), and test set accuracy, for an increasing number of perturbations. **Left to Right:** HELOC, German Credit, and Adult Income datasets, with noise added to first layer biases, weights, and weights, respectively. Observe how explanation similarity may increase, even as test accuracy drops (one is not a strong indicator of the other). Error bars represent the central decile of SA scores over 1000 individuals, and the interquartile ranges of accuracies over perturbed models.

Figure 5: Effect of $\sigma$ on top-k SA scores (for input gradients and $k = 5$), and test set accuracy, over different layers perturbed 100 times. **Left to Right:** HELOC, Default Credit, and GMSC datasets. Error bars represent the central decile of SA scores over 1000 individuals, and the interquartile ranges of accuracies over perturbed models.

**Mode connectivity implementation** We implement mode connectivity following the method outlined in Garipov et al. (2018). This process involves training a curve between two endpoints, which, while beneficial, introduces additional computational costs. An alternative is to train the two endpoints from scratch, using identical hyperparameters to those used for individual models. This approach requires the same computational effort as independently training two models, though is often faster due to internal parallelization. The loss for each batch is calculated from a uniformly random point sampled along the curve, and the weights of the endpoints are then updated via backpropagation. We confirm that this results in endpoint weights closely aligned with those trained independently, serving as an effective approximation of the mode connectivity process, and detail our approach in Appendix C. Future work should strive to further explore properties of the loss landscape in order to expedite mode connectivity, building upon recent advancements in the field (Ainsworth et al., 2023; Gotmare et al., 2018; Singh and Jaggi, 2020; Tatro et al., 2020).

### 4.1. Impact of Weight Perturbation

We begin by performing ablations over the value of $\sigma$ used for weight perturbation, sampling 24 models without replacement from the underspecification set of 1000 pretrained models. Each model is perturbed a number of times, yielding 24 *perturbed models* (a form of ensemble gener-

ated from just one pre-trained model, described in §3.1). We measure SA similarity of top-5 gradient features between all $\binom{24}{2} = 276$ unique pairs of perturbed models, averaging over all pairs, i.e. error bars correspond to individuals in the test set. Test accuracy is computed using *all* test inputs, with errors bars corresponding to the perturbed models.

These experiments aim to ascertain the impact of local weight perturbations on explanation similarity and test accuracy, thereby informing the optimal setup for our method. Figure 4 indicates that perturbing the weights or biases of the first layer with Gaussian noise $\epsilon \sim \mathcal{N}(0, \sigma^2)$ can lead to significant increases in explanation similarity (measure between perturbed models). For instance, over values of $\sigma$ where test accuracy remains approximately constant, SA scores on HELOC increase from around 0.62 to 0.74, by perturbing just 20 times the biases of the first layer (in general, perturbing biases converged faster compared to weights). These gains correspond to perturbing one model. The cumulative effect of constructing ensembles from multiple perturbed models, discussed in §4.2, leads to further gains.

We also investigate separately perturbing each layer of the network (Figure 5), finding that explanation similarity *depends critically on the layer being perturbed*[3]. We observe

---

[3]For clarity, our naming convention is that *Layer 1* refers to weights/biases between input and first hidden layer, *Layer 2* refers to weights/biases between first and second hidden layer, etc.

*Gradients Average Pairwise Signed-Set Agreement (SSA) between Ensembles vs Number of Pretrained Models per Ensemble*

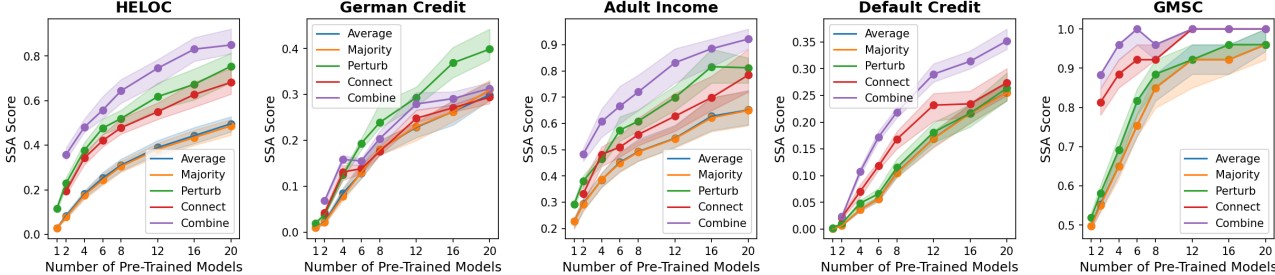

Figure 6: Effectiveness of ensemble strategies in stabilizing explanations (top-5 SSA scores, input gradients, all datasets). Observe the increases in similarity between ensembles when combining global (mode connectivity) and local (perturbation) explorations. *Average* and *Majority* denote vanilla ensembles. *Perturb*, *Connect*, and *Combine* denote weight perturbation, mode connectivity, and their combination.

that the weights or biases of the first layer tend to be most sensitive to perturbations, providing potential for improved explanation similarity. For instance, SA scores do not significantly increase in Figure 5 when perturbing weights/biases beyond the first layer. Further investigations, including full ablations across all datasets, are located in Appendix B.

### 4.2. Comparison of Ensemble Techniques

We construct ensembles by randomly sampling 50 non-overlapping sets of models from the underspecification set (1000 pre-trained models). The size of each set differs per experiment, with a maximum size of 20 pre-trained models (covering all models). For a given $k$ value and explanation technique, we compute similarity scores between all $\binom{50}{2}$ = 1225 unique pairs of ensembles, averaging over all pairs. Thus, scores represent the range of probability values that individuals have of satisfying certain criteria. For example, SSA scores estimate the likelihood that individuals maintain identical top-$k$ features (ignoring ordering), when switching between approximately equivalent models.

We also trial the combination of both weight perturbation (*perturb*) and mode connectivity (*connect*), as a novel ensembling technique (denoted *combine*), which uniformly samples models along each mode connecting path, and perturbs each of them. Results across all datasets, explanation methods, and similarity metrics are in Appendix D.

**Vanilla ensembles can eliminate explanatory multiplicity, when sufficiently large** Whether the distribution of model explanations across the underspecification set has low or high variance, the explanations between vanilla ensembles of sufficient size sampled from the set will eventually converge. We observe this convergence in Figure 6. While vanilla ensembles of just 20 pre-trained models cannot achieve perfect similarity in all cases, as the ensembles grow in size we see a steady increase in average pairwise SSA score (the strictest criterion). For GMSC, and vanilla

ensembles (blue and orange), the median probability of an individual retaining the exact same top-$k$ features when switching between models increases from 50% to 95%.

**Strategic ensembling reduces pre-training requirements** We demonstrate in Figure 6 that our ensemble strategies consistently expedite the alignment of model explanations i.e., require fewer pre-trained models to achieve a given similarity score. The relative performance gains that we achieve through either local (green) or global (red) exploration often hinge upon the unique characteristics of the given dataset, explanation technique, or model class used. We observe that our approach of combining both forms of exploration achieves superior performance in most cases. Notably, to increase SSA scores to around 40% in HELOC, from 0% for single models, vanilla ensembles required 12 pre-trained models (blue and orange). Conversely, the combined exploration of mode connectivity and weight perturbation required just 2 pre-trained models (purple). Similarly, the number of pre-trained models required to surpass SSA scores of 60% on Adult Income, can be reduced fivefold from 20 to just 4. Table 1 showcases further examples in which our methods comprehensively improve explanation similarity across SA, SSA and CDC metrics, for ensembles constructed with four pre-trained models. Note that the CDC metric typically returned 1 for top-5 explanations, and is the least strict of the three metrics. We instead consider top-$d$, computing CDC over all $d$ features in each dataset.

**Ensembling increases test accuracy** A well known property of ensembles is their ability to reduce generalization error (Dieterich, 2000; Allen-Zhu and Li, 2023), and we observe the same results in our experiments. Our value of $\sigma$ was chosen to maximize similarity, but with the constraint that the test accuracy does not drop by more than 1%. In most cases, our ensembling methods achieved similar test accuracies to vanilla ensembles, with test accuracy exceeding standard ensembles by up to 1% in some cases.

Table 1: Results of explanation alignment for Smoothgrad and DeepSHAP, across all datasets and similarity metrics. Ensembles are constructed from just four pre-trained models. Explanation agreement between *single* models sampled from the underspecification set are shown for reference. We report mean values alongside standard deviation. Best values are shown in bold.

| Dataset | Ensemble Method | Smoothgrad | | | DeepSHAP | | |
|---|---|---|---|---|---|---|---|
| | | Top-5 SA | Top-5 SSA | Top-d CDC | Top-5 SA | Top-5 SSA | Top-d CDC |
| **HELOC** (23 features) | Single | 0.65 ± 0.09 | 0.06 ± 0.06 | 0.01 ± 0.01 | 0.75 ± 0.09 | 0.18 ± 0.17 | 0.02 ± 0.02 |
| | Vanilla | 0.81 ± 0.08 | 0.25 ± 0.19 | 0.09 ± 0.09 | 0.84 ± 0.09 | 0.39 ± 0.27 | 0.09 ± 0.09 |
| | Perturb | 0.87 ± 0.06 | 0.43 ± 0.23 | 0.19 ± 0.14 | 0.86 ± 0.07 | 0.44 ± 0.23 | 0.12 ± 0.08 |
| | Connect | 0.86 ± 0.07 | 0.39 ± 0.22 | 0.15 ± 0.13 | 0.88 ± 0.08 | 0.51 ± 0.28 | 0.11 ± 0.10 |
| | Combine | **0.90 ± 0.06** | **0.55 ± 0.25** | **0.26 ± 0.17** | **0.90 ± 0.07** | **0.55 ± 0.28** | **0.14 ± 0.10** |
| **German Credit** (70 features) | Single | 0.50 ± 0.08 | 0.01 ± 0.01 | 0.00 ± 0.00 | 0.68 ± 0.11 | 0.11 ± 0.15 | 0.00 ± 0.00 |
| | Vanilla | 0.69 ± 0.08 | 0.09 ± 0.07 | 0.00 ± 0.00 | 0.80 ± 0.09 | 0.27 ± 0.24 | 0.00 ± 0.00 |
| | Perturb | 0.73 ± 0.07 | 0.12 ± 0.08 | 0.00 ± 0.00 | **0.83 ± 0.08** | **0.32 ± 0.22** | 0.00 ± 0.00 |
| | Connect | 0.71 ± 0.12 | 0.12 ± 0.10 | 0.00 ± 0.00 | 0.77 ± 0.10 | 0.22 ± 0.21 | 0.00 ± 0.00 |
| | Combine | **0.76 ± 0.08** | **0.17 ± 0.12** | 0.00 ± 0.00 | 0.81 ± 0.08 | 0.27 ± 0.21 | 0.00 ± 0.00 |
| **Adult Income** (6 features) | Single | 0.83 ± 0.07 | 0.29 ± 0.18 | 0.32 ± 0.15 | 0.92 ± 0.07 | 0.63 ± 0.31 | 0.51 ± 0.17 |
| | Vanilla | 0.89 ± 0.06 | 0.48 ± 0.24 | 0.50 ± 0.19 | 0.95 ± 0.06 | 0.75 ± 0.26 | 0.66 ± 0.20 |
| | Perturb | 0.90 ± 0.05 | 0.54 ± 0.24 | 0.54 ± 0.19 | 0.95 ± 0.05 | 0.77 ± 0.23 | 0.77 ± 0.19 |
| | Connect | 0.90 ± 0.07 | 0.53 ± 0.27 | 0.53 ± 0.22 | 0.96 ± 0.05 | 0.79 ± 0.24 | 0.74 ± 0.20 |
| | Combine | **0.92 ± 0.06** | **0.61 ± 0.26** | **0.58 ± 0.21** | **0.96 ± 0.04** | **0.81 ± 0.21** | **0.78 ± 0.20** |
| **Default Credit** (90 features) | Single | 0.49 ± 0.06 | 0.00 ± 0.00 | 0.00 ± 0.00 | 0.82 ± 0.07 | 0.29 ± 0.18 | 0.00 ± 0.00 |
| | Vanilla | 0.67 ± 0.07 | 0.04 ± 0.04 | 0.01 ± 0.02 | 0.89 ± 0.07 | 0.54 ± 0.27 | 0.02 ± 0.04 |
| | Perturb | 0.72 ± 0.06 | 0.07 ± 0.07 | 0.02 ± 0.03 | 0.90 ± 0.07 | 0.54 ± 0.27 | 0.03 ± 0.04 |
| | Connect | 0.70 ± 0.07 | 0.07 ± 0.05 | 0.02 ± 0.03 | 0.91 ± 0.06 | 0.58 ± 0.25 | 0.03 ± 0.03 |
| | Combine | **0.75 ± 0.05** | **0.11 ± 0.05** | **0.03 ± 0.06** | 0.91 ± 0.06 | 0.58 ± 0.25 | **0.03 ± 0.02** |
| **GMSC** (10 features) | Single | 0.91 ± 0.03 | 0.57 ± 0.15 | 0.46 ± 0.23 | 0.85 ± 0.07 | 0.39 ± 0.21 | 0.39 ± 0.19 |
| | Vanilla | 0.95 ± 0.04 | 0.75 ± 0.19 | 0.67 ± 0.25 | 0.90 ± 0.07 | 0.60 ± 0.25 | 0.58 ± 0.22 |
| | Perturb | 0.95 ± 0.04 | 0.76 ± 0.19 | 0.69 ± 0.24 | 0.90 ± 0.07 | 0.60 ± 0.25 | 0.59 ± 0.21 |
| | Connect | 0.96 ± 0.04 | 0.82 ± 0.18 | 0.71 ± 0.25 | 0.90 ± 0.06 | 0.58 ± 0.25 | 0.62 ± 0.20 |
| | Combine | **0.97 ± 0.04** | **0.83 ± 0.18** | **0.72 ± 0.24** | **0.91 ± 0.06** | **0.62 ± 0.26** | **0.66 ± 0.21** |

## 5. Limitations & Future Work

Our work, while providing significant insights into the behaviour of model explanations, acknowledges certain limitations. Notably, the selection of a single optimal hyperparameter set for each dataset, despite multiple sets offering a potential breadth of solutions, is a potential direction for future work. Comparing explanations across multiple underspecification sets may present additional challenges but also provide further insights into the behavior of explanations.

Additionally, the methods we provide traverse the loss landscape along two fundamental axes: locally (with perturbations) and globally (between models). There is much to learn and optimize about these methods. Future work would include refining the methods of weight perturbation, and exploring alternate paths for mode connectivity (Gotmare et al., 2018; Zhao et al., 2020), in order to develop better heuristics for navigating the loss landscape. The ultimate aim is to devise more effective strategies for ensemble creation that balance the goals of model performance, explanation consistency, and computational efficiency.

Finally, although our methods require no extra training compared to standard ensembling, approaches to reduce the total number of models included in the set should be explored to cut inference costs. This could involve techniques such as aligning constituent models in weight space through permutation symmetries (Ainsworth et al., 2023; Singh and Jaggi, 2020; Tatro et al., 2020), or investigating methods that find a single weight configuration matching the output of en-

sembles, to reap the benefits of ensembling, while reducing computational costs associated with operating large numbers of constituent models. In this case, examining the effects of distillation (Hinton et al., 2015) and self-distillation (Furlanello et al., 2018) could provide additional insights.

A more comprehensive discussion on the limitations and prospective avenues of study, including the exploration of alternate explanation methods, are detailed in Appendix E.

## 6. Conclusion

Here, we tackle the challenge of providing consistent explanations for predictive models in the presence of model indeterminacy from the underspecification set. Motivated by local and mode-connected loss landscape exploration, we develop two novel ensembling methods. On five benchmark financial datasets, our methods markedly increase the consistency of model explanations when using a fixed number of pre-trained models. Moreover, our methods are computationally more affordable than standard ensembling techniques with respect to the number of trained model constituents, while not compromising on test accuracy.

It is important to acknowledge that our work does not explicitly consider the broader impacts and desirable properties of these ensembles in application, such as for fairness or bias. While our work serves as a foundation for exploring the impact of ensembling on explanation consistency, we encourage further research into how our techniques may interact with other aspects of AI safety.

**Acknowledgments:** The authors would like to thank the anonymous reviewers for helpful feedback and all the funding agencies listed below for supporting this work. This work is supported in part by the NSF awards #IIS2008461, #IIS-2040989, and #IIS-2238714, and research awards from Google, JP Morgan, Amazon, Harvard Data Science Initiative, and D^3 Institute at Harvard. The views expressed here are those of the authors and do not reflect the official policy or position of the funding agencies.

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

# Appendix

This appendix is formatted as follows.

1. We discuss the *Datasets & Models* used in our work in Appendix A.

2. We discuss *Weight Perturbation* and display full results from ablations in Appendix B.

3. We detail our *Mode Connectivity* implementation in Appendix C.

4. We display full *Comparison of Ensemble Techniques* in Appendix D.

5. We discuss *Limitations & Future Work* in Appendix E.

Where necessary, we discuss potential limitations of our work and future avenues for exploration.

## A. Datasets & Models

Five benchmark datasets are employed in our experiments, all of which describe binary classification and are publicly available. Details are provided in Appendix A.1 and in Table 2. Our experiments are conducted for a fixed architecture, trained with hyperparameters determined by model selection sweeps. Details are provided in Appendix A.2 and Table 3.

### A.1. Datasets

The **HELOC** (Home Equity Line of Credit) dataset (FICO, 2018) classifies **credit risk** and can be obtained from (upon request) and is described in detail at: `https://community.fico.com/s/explainable-machine-learning-challenge`. We drop duplicate inputs, and inputs where all feature values are missing (negative), and replace remaining missing values in the dataset with median values.

The **German Credit** dataset (Dua and Graff, 2017) classifies **credit risk** and can be obtained from and is described at: `https://archive.ics.uci.edu/ml/datasets/statlog+(german+credit+data)`. The documentation for this dataset also details a cost matrix, where false positive predictions induce a higher cost than false negative predictions, but we ignore this in model training. Note that this dataset includes mostly categorical features.

The **Adult Income** dataset (Dua and Graff, 2017) classifies whether an individual's **salary** exceeds $50,000, and is obtained from and described at: `https://archive.ics.uci.edu/ml/datasets/adult`. We drop categorical features for this dataset, resulting in 6 features total, to include an assessment of a case where *top-5* would encompass most features (n.b., we still see disagreement in this case- either the signs of the 5 features disagreed, or the least important $6^{th}$ feature disagreed).

The **Default Credit** dataset (Yeh and Lien, 2009) classifies **default risk** on customer payments and is obtained from and described at: `https://archive.ics.uci.edu/ml/datasets/default+of+credit+card+clients`. This dataset, and German Credit, stress-test increased dimensions (number of features).

The **GMSC** (Give Me Some Credit) dataset (Freshcorn, 2022) classifies the **probability of default** within the next two years, and is obtained from and described at: `https://www.kaggle.com/datasets/brycecf/give-me-some-credit-dataset`. We pre-process a large random sample of this dataset, ensuring a 50-50 split between class 0 (no default) and class 1 (default). Summary of these datasets is in Table 2.

Table 2: Summary of the datasets used in our experiments.

| Name | No. Inputs | Input Dim. | Categorical | Continuous | No. Train | No. Test |
|---|---|---|---|---|---|---|
| HELOC | 9871 | 23 | 0 | 23 | 7896* | 1975* |
| German Credit | 999 | 70* | 17 | 3 | 799 | 200 |
| Adult Income | 32561 | 6 | 0 | 6 | 26048 | 6513 |
| Default Credit | 30000 | 90* | 9 | 14 | 24000 | 6000 |
| GMSC | 11426 | 10 | 0 | 10 | 9140 | 2286 |

*Denotes values post-processing (one-hot encoding inputs, dropping inputs).

Table 3: Hyperparameters used to train our models, and model performance.

| Name | Epochs | Learning Rate | Batch Size | Train Acc. (%) | Test Acc. (%) |
|---|---|---|---|---|---|
| HELOC | 20 | 0.0004 | 32 | $77.97 \pm 0.31$ | $72.90 \pm 0.45$ |
| German Credit | 100 | 0.004 | 32 | $92.81 \pm 1.98$ | $73.47 \pm 1.47$ |
| Adult Income | 40 | 0.004 | 128 | $83.50 \pm 0.15$ | $82.12 \pm 0.16$ |
| Default Credit | 10 | 0.0001 | 128 | $82.32 \pm 0.05$ | $82.67 \pm 0.09$ |
| GMSC | 20 | 0.0004 | 32 | $75.17 \pm 0.34$ | $73.96 \pm 0.34$ |

## A.2. Models

We use `PyTorch` (Paszke et al., 2019) to implement feedforward neural networks with fixed architectures: hidden layers of size 128, 64, and 16. We use ReLU activations for the intermediate layers and Softmax for the output. Model selection sweeps are performed with `RayTune` (Liaw et al., 2018), to identify the best performing model hyperparameters for a given dataset (shown in Table 3). We performed iterative searches across learning rates from $10^{-6}$ to $10^{-1}$, batch sizes 16 to 128, and epochs typically between 5 and 100 (larger values only where necessary). We employ a filtering process to discard models that perform >1% below the mean accuracy, though recognize the importance of seed-induced variability in the context of these systems. We initially tested both filtering and not filtering, finding the overall results to be largely similar.

## B. Weight Perturbation

We present here full ablations across all datasets, to identify the optimal layer to perturb and the corresponding standard deviation $\sigma$ and number of perturbations, depicted in Figures 7 and 8. As previously noted, the effects of random noise perturbations depend critically on the layer being perturbed. In most cases (bar German Credit or Adult Income), perturbing layers deeper than the first resulted in no increases in explanation similarity, and optimal gains were found by perturbing either the weights or biases of the first layer (i.e. connections between the input and the first hidden layer of size 128).

In our ensemble experiments, we perturb first layer biases with $\sigma = 0.05, 0.5$, for GMSC and Default Credit, respectively. We perturb first layer weights with $\sigma = 0.2, 0.2, 0.3$ for HELOC, German Credit and Adult Income, respectively. We use 50 perturbations in all cases, besides HELOC where we use 100 perturbations. We also trialled perturbing layers cumulatively, or perturbing layers with noise proportional to the loss gradient of each weight, though could not identify superior results.

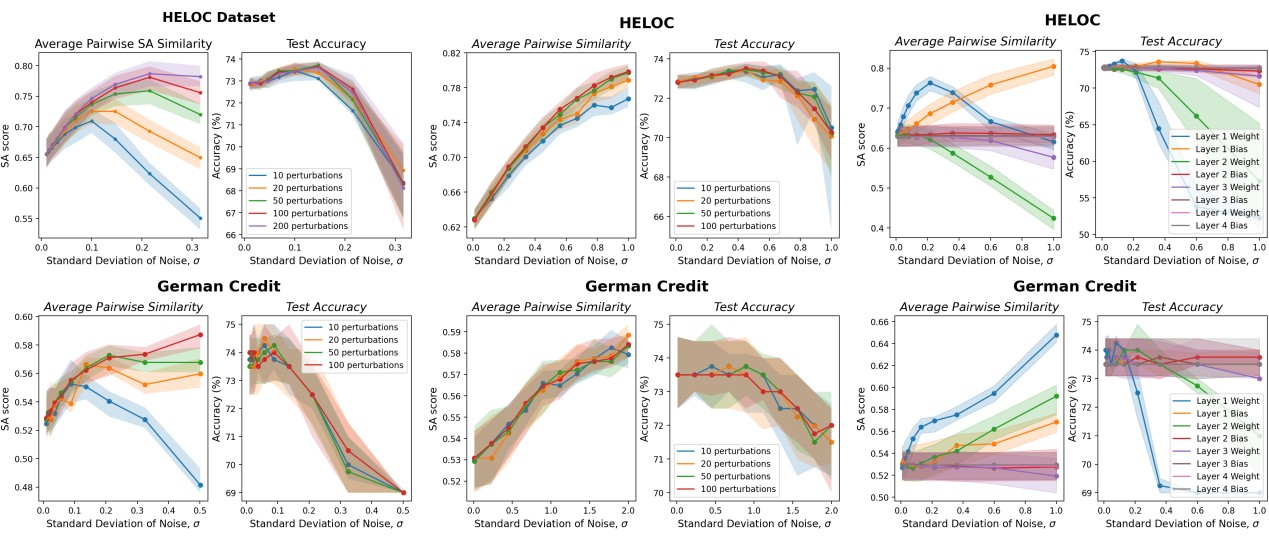

Figure 7: Effect of $\sigma$ on top-k SA scores (input gradients, $k = 5$) and test accuracy on HELOC and German Credit datasets. **Left and Center:** perturbing the first layer weights, and the first layer biases, respectively. **Right:** perturbing layers individually 100 times each. Error bars represent the central decile of SA scores over 1000 individuals, and the interquartile range of accuracies over perturbed models.

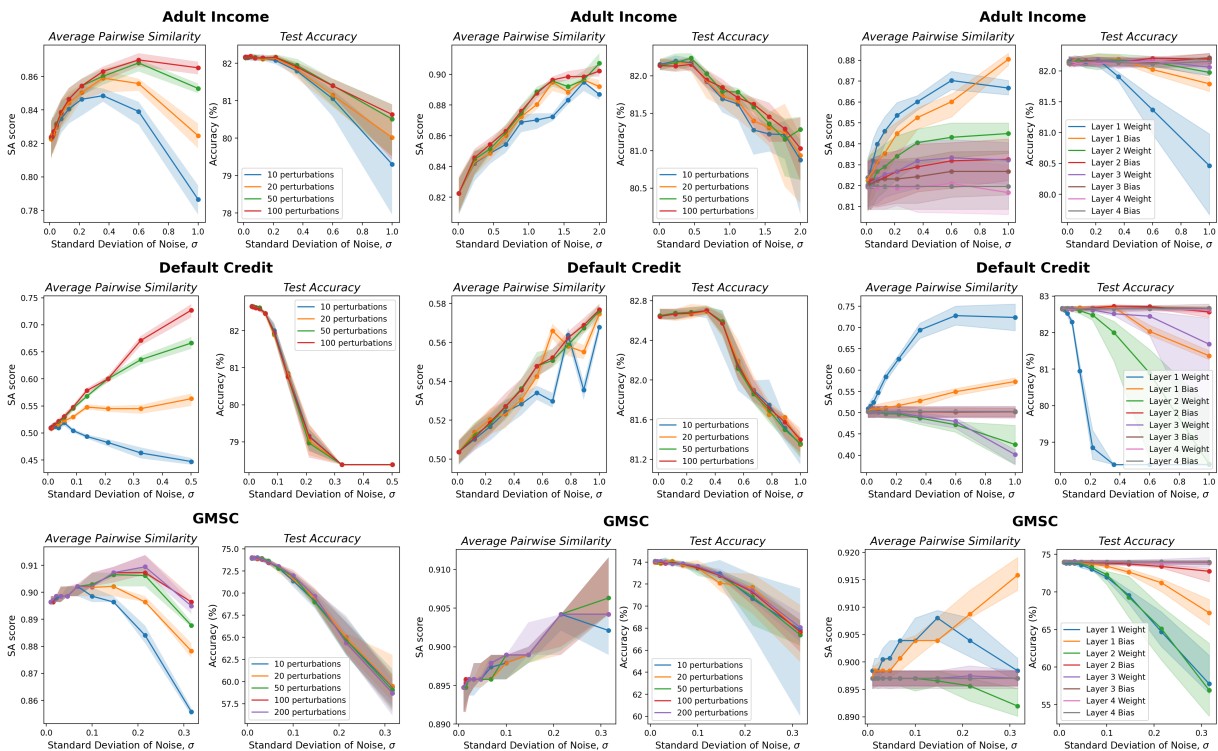

Figure 8: Effect of $\sigma$ on top-k SA scores (input gradients, $k = 5$) and test accuracy on Adult Income, Default Credit and GMSC. **Left and Center:** perturbing first layer weights, and first layer biases, respectively. **Right:** perturbing layers individually 100 times each. Error bars represent the central decile of SA scores over 1000 individuals, and the interquartile range of accuracies over perturbed models.

## C. Mode Connectivity

This section details our mode connectivity implementation. Given two models from the underspecification set, one can connect the two models in weight space via paths or near constant loss. We use the publicly available code from Garipov et al. (2018) to do so. This takes two models as a fixed startpoint and endpoint for the curve. A linear path between the startpoint and endpoint is initialized, and a fixed number of bends lying along this path are backpropagated in order to minimize training loss of points sampled uniformly randomly along the curve.

While this method yielded similar improvements, it introduces additional training costs, since one must effectively train new models (the bends along the curve). An alternate approach, described in the main text, is to initialize the curve with untrained start and end points, and train the curve from scratch. We trialled this as an effective approximation to mode connecting process. Specifically, the startpoint of one curve is initialized with the same random seed as used to train a given

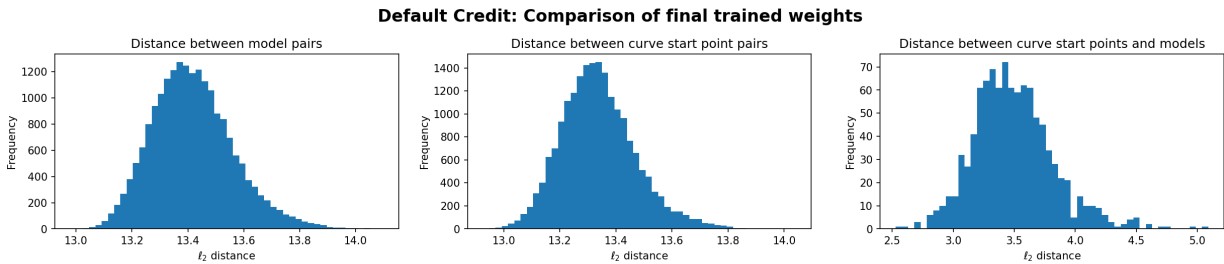

Figure 9: Mode connectivity approximation for Default Credit. Curve startpoints are similarly positioned in weight space to the original models, after training curves from scratch. **Left:** $\ell_2$ distance in weight space between model pairs in the underspecification set. **Center:** $\ell_2$ distance in weight space between curve *startpoint* pairs. There is similar distance between pairs of curve startpoints after training curves, compared to between standard model pairs. **Right:** $\ell_2$ distance between trained curve startpoints and trained models is low.

model from the underspecification set (endpoint is initialized randomly). The curve is then trained with the exact same hyperparameters as the original model was trained with. The only difference is that loss is sampled along the curve, rather than just at the startpoint. We demonstrate in Figure 9 how this resulted in the startpoint of the curve being trained to a similar position in weight space as the original model (i.e. $\ell_2$ distances are low on the right hand graph, while distances between pairs of curve startpoints after training is similar to distances between pairs of models after training).

Future work can consider optimizing the mode connectivity process, to find paths with reduced computational cost. Multiple prior work have indicated that *linear mode connectivity*, i.e. straight-line paths between models of near constant loss, can be achieved via permutation of the weights used (Ainsworth et al., 2023; Singh and Jaggi, 2020; Tatro et al., 2020). It is worth noting that the focus of our investigation was to analyze how *local*, *global*, or their *combined* explorations of the loss landscape could promote explanation alignment after ensembling. The exact mechanisms used to explore the loss landscape, and their computational efficiencies, should build on related advancements, and is a ripe area for future work.

## D. Comparison of Ensemble Techniques

This Appendix displays full results, similar to those in Figure 6, but across all similarity metrics and explanation methods. Recall that for the CDC metric, we aim to stress test our ensembling techniques by assessing top-$d$ similarity, where $d$ is the dimensionality of the data, i.e. a score of 1 is returned only if the signs of all features match between two ensembles. For a given number of pre-trained models, 50 ensembles are constructed, and for each test point, average similarities are computed across all $\binom{50}{2} = 1225$ unique pairs. The median similarity is plotted (error bars indicate the central decile of individuals).

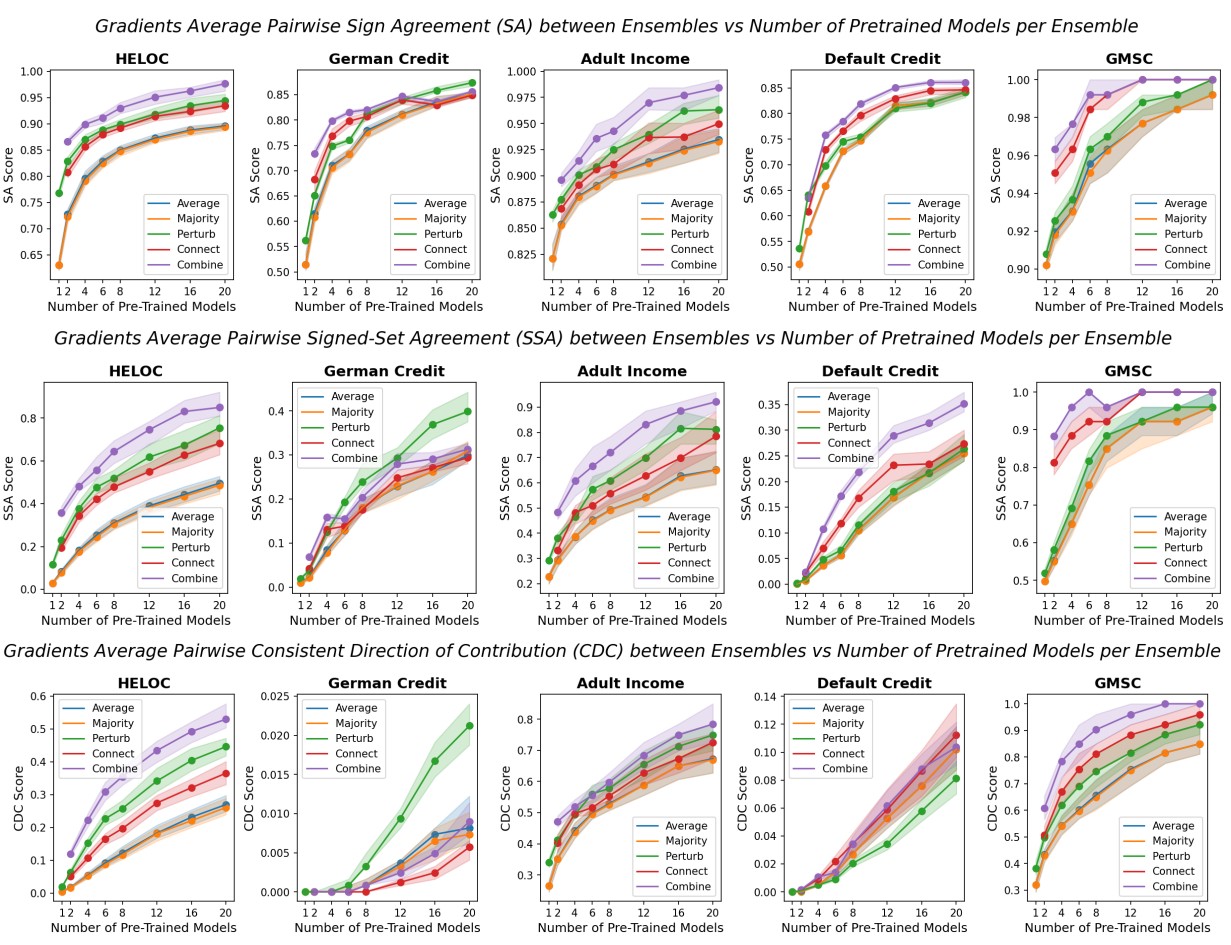

Figure 10: Effectiveness of ensemble strategies in stabilizing explanations across all datasets (top-5 SA, top-5 SSA, and top-$d$ CDC scores for **input gradients**). *Average* and *Majority* denote vanilla ensembles, while *Perturb*, *Connect*, and *Combine* denote weight perturbation, mode connectivity, and their combination respectively.

**Effects of ensembling on Saliency** Figure 10 details full results for input gradients i.e. Saliency. The central row is depicted in the main text (Figure 6). Observe similar trends between SA scores (top row) and SSA scores (central row), as the latter is a stricter (binary) version of the former. Top-$d$ CDC comparisons demonstrate similar trends. For instance, to exceed 80% similarity on the GMSC dataset, standard ensembling (blue and orange) requires around 20 pre-trained models, while the combined exploration of mode connectivity alongside local perturbations (purple) requires 4 pre-trained models.

**Effects of ensembling on Smoothgrad** Figure 11 details full results for Smoothgrad, where we use 50 perturbations on the input with $\sigma = 0.1$. Note that this explanation technique generally improves explanation similarity slightly for standard ensembles (blue and orange), e.g., for 20 pre-trained models, median SSA scores on Adult Income increase from 60% in Figure 10 to 70% in Figure 11. However, our findings suggest that Smoothgrad does not provide significant smoothing benefits for explanation alignment beyond our ensemble techniques. Specifically, for the same comparison of median SSA scores on Adult Income, when considering 20 pre-trained models, both Smoothgrad and saliency techniques achieve approximately 90% explanation alignment using the combined approach, while local perturbations or mode connectivity yield around 80% alignment. Furthermore, while Smoothgrad provides an explanation for a smoothed approximation of a potentially noisy decision boundary, ensemble approaches smooth both the explanation and the model's decision region.

**Effects of ensembling on DeepSHAP** Figure 12 details full results for DeepSHAP, an approximation to SHAP that leverages knowledge from the neural network directly. Note first that results may be slightly noisier, given the computational restriction of using this method (the first 100 test points in each datasets are evaluated, rather than the first 1000 as in other methods). In spite of this, the various ensembling techniques proposed, for a given number of pre-trained models, tend

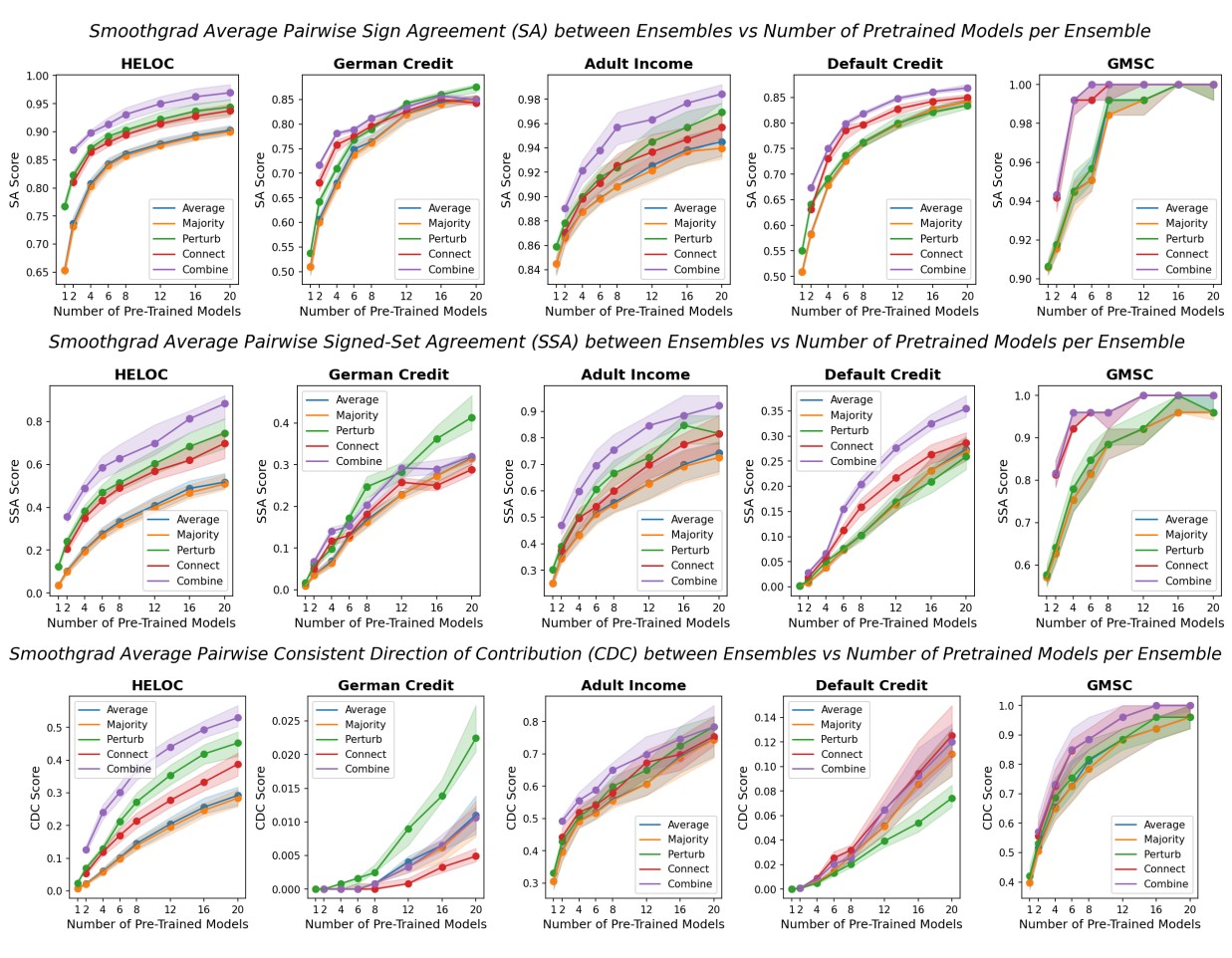

Figure 11: Effectiveness of ensemble strategies in stabilizing explanations across all datasets (top-5 SA, top-5 SSA, and top-$d$ CDC scores for **Smoothgrad**). *Average* and *Majority* denote vanilla ensembles, while *Perturb*, *Connect*, and *Combine* denote weight perturbation, mode connectivity, and their combination respectively.

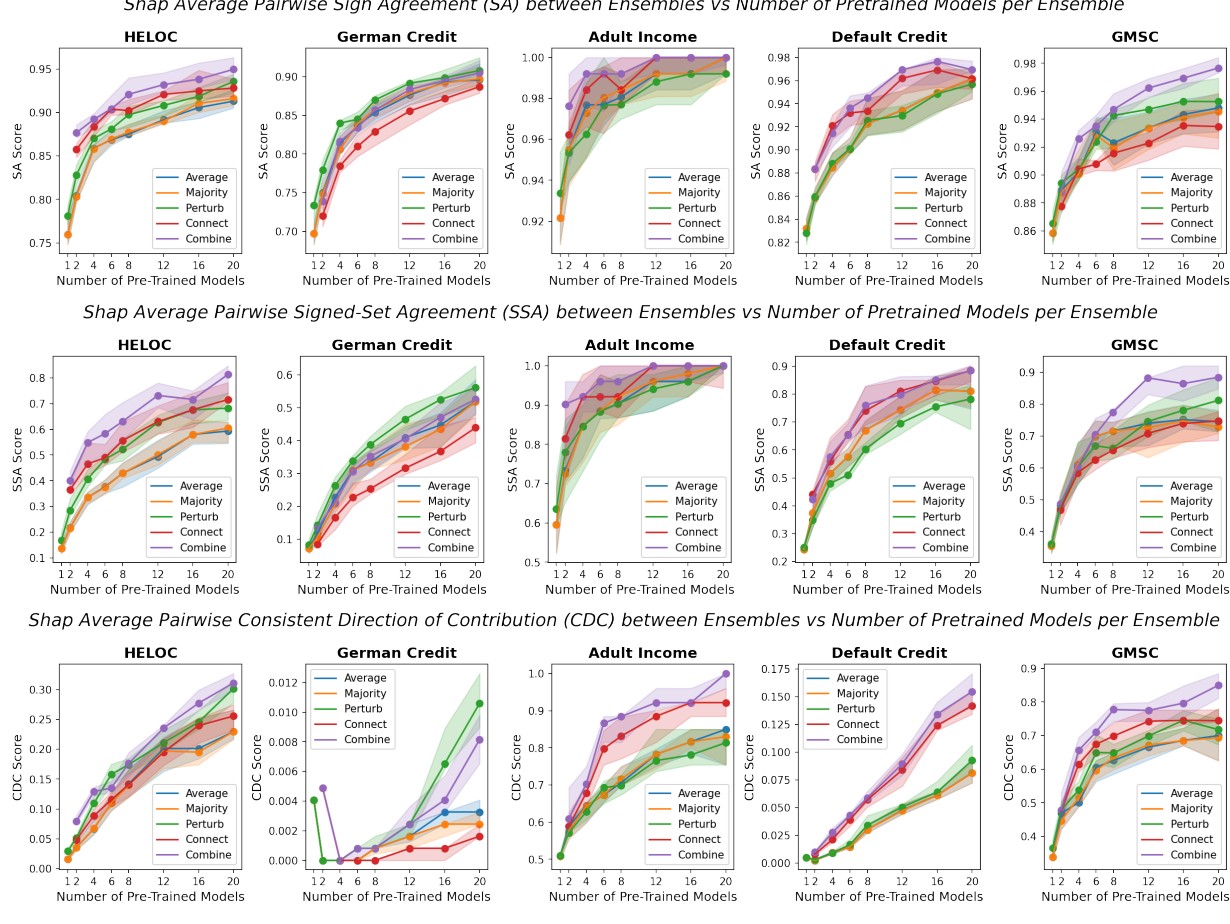

Figure 12: Effectiveness of ensemble strategies in stabilizing explanations across all datasets (top-5 SA, top-5 SSA, and top-$d$ CDC scores for **DeepSHAP**). *Average* and *Majority* denote vanilla ensembles, while *Perturb*, *Connect*, and *Combine* denote weight perturbation, mode connectivity, and their combination respectively.

to yield benefits similar to previously e.g. HELOC trends are largely similar after accounting for noise fluctuations. SA and SSA scores for DeepSHAP tended to have higher baseline similarity scores (i.e. between single models) compared to Saliency and Smoothgrad. We see notable improvements for the mode connected/combined explorations for Adult Income, Default Credit and GMSC. However, we also observe the one instance where mode connectivity was inferior to standard ensembling, in German Credit. While this was overcome through the combined technique of sampling along connected modes and further perturbing each of the samples (purple), and secondly that local weight perturbation alone improved performance, this emphasizes the unpredictable nature of particular explanation techniques, and warrants future research. A *sufficiently large* number of pre-trained models may eliminate explanation multiplicity, and so the focus of future work should be on exploiting further the properties of the loss landscape to optimize the search for ensemble members.

# E. Limitations & Future Work

This work presents a novel approach to enhancing the consistency of model explanations by leveraging ensembling methods based on loss landscape exploration. While we observe promising results, we acknowledge that several interesting avenues for future work emerge from our study.

**Underspecification set**     In this work, we define the underspecification set as the collection of optimal models trained with a fixed set of hyperparameters, with the only source of variation due to the random seed using in training. By focusing on the underspecification set, we aim to shed light on the intrinsic inconsistency of model explanations arising purely from indeterminacy within a specific model configuration, rather than across different configurations.

However, we acknowledge that in real-world scenarios, an exact underspecification set may not always be found. Often, there are multiple sets of near-optimal hyperparameters, each potentially giving rise to a different underspecification set. The focus on one such set in our study is intentional, as it provides a clearer landscape for assessing the effectiveness of ensemble techniques in handling model indeterminacy. It is important to contextualize that addressing the complexities within the underspecification set is a necessary first step. Without a sound understanding of the explanatory behavior within a specific model configuration, attempting to reconcile explanations across broader ranges – such as the entire Rashomon set encompassing different model classes and hyperparameters – would be overly ambitious. Thus, our focus on the underspecification set offers an essential foundation for further research in the quest for more consistent model explanations.

Looking forward, an interesting direction for future work is the exploration of multiple underspecification sets concurrently. Comparing and consolidating explanations across multiple underspecification sets may present additional challenges but also provide further insights into the behavior of model explanations. This exploration would segue naturally into investigating the Rashomon set, to capture the variation across all possible well-performing models.

**Alternate explanation methods**   We have chosen to use gradient-based methods for generating explanations in this study due to their simplicity, computational efficiency, and intuitive appeal. The gradient of a model with respect to its inputs can provide valuable insights about how the model responds to changes in those inputs. In essence, it can inform us about the local sensitivity of the model's predictions and thus can be interpreted as a form of local explanation method. Moreover, gradients can be seen as a natural proxy for counterfactual explanations (CEs), another popular category of explanation methods. CEs identify the minimal changes required in the input features to achieve a different prediction. Because gradients indicate the direction of the greatest change in model output, they can be seen as a first approximation to CE directions.

However, it is worth noting that CEs, while providing a powerful intuitive appeal, come with their own set of challenges. They often require more computation than gradient-based methods, and they can be sensitive to the specific definition of "minimal change", which can depend on domain-specific factors. While we have focused on gradient-based explanations for their straightforwardness and direct relation to counterfactual reasoning, the field is open for further investigation and exploration of alternative methods for enhancing the consistency of model explanations.

We acknowledge that there is a wide array of explanation methods available, each with its unique advantages and limitations, and many of which could be considered in the context of our framework. Future work could explore other types of explanations, including counterfactual methods, or prototype-based methods, among others. Each of these could provide different perspectives on the consistency of model explanations and their susceptibility to model indeterminacy.

**Weight perturbations and mode connectivity**   We employ two strategies for navigating the loss landscape: local exploration via weight perturbations and global exploration via mode connectivity. Weight perturbations could be perceived as the 'safer' strategy, presenting a lower risk profile (dependent on the standard deviation of perturbations) but potentially hitting a performance ceiling based on dataset and model class (Section 4.1 and Appendix B). The performance of this method can be further optimized by considering a selection process for the perturbations, possibly based on training accuracy, or by adjusting the sampling strategy for the perturbation magnitude, taking into account the hyperspherical distribution of samples in high-dimensional weight space. On the other hand, mode connectivity may be perceived as 'riskier' due to its broader scope but could yield benefits that go beyond the reach of simple local exploration, allowing us to explore more distant regions of the loss landscape, connecting different locally optimal solutions.

Looking ahead, there is much to learn and optimize about these methods. Future work would include both refining the methods of weight perturbation, and exploring alternate paths for mode connectivity (Ainsworth et al., 2023; Gotmare et al., 2018; Singh and Jaggi, 2020; Tatro et al., 2020; Zhao et al., 2020), in order to develop better heuristics for navigating the loss landscape. The ultimate aim is to devise more effective strategies for ensemble creation that balance the goals of model performance, explanation consistency, and computational efficiency.

**Improving inference efficiency**   Although our methods require no extra training compared to standard ensembling, approaches to reduce the total number of models included in the set should be explored to cut inference costs.[4]   For instance, using ensembles with 10 pre-trained models each perturbed 50 times totals 500 models. While our methods offer computational efficiency compared to standard ensembling techniques with respect to the number of pre-trained models required, the practicality of this approach might still be challenging for larger scale applications. Future work could focus on

---

[4]Note additionally that our implementations may not be fully optimized with respect to parallelization, etc.

finding efficient ways to reduce the total number of models in the ensemble. This could involve techniques such as weight permutations to align models and subsequently averaging the weights, or investigating methods to find a single weight configuration that matches the output of ensembles, which we detail in the following two paragraphs.

**Permutation symmetries**    The initial investigations provided in this work could greatly benefit from the exploration of permutation symmetries. This emerging direction in research promises a more efficient traversal of the underspecification set (Ainsworth et al., 2023; Singh and Jaggi, 2020; Tatro et al., 2020). The strategy involves aligning constituent models in weight space through permutation symmetries, thereby reducing the complexity of exploring numerous paths in the loss landscape. Through this approach, we hope to optimize the exploration process, enabling faster and more effective generation of ensembles. Not only would this strategy potentially reduce computational demands, it may also lead to uncovering more consistent explanations across ensembles, and aid in enhancing our understanding of model indeterminacy.

**Consolidating ensemble models**    The concept of consolidating ensembles into a single point in weight space stands as another promising direction for future work. The intention behind this proposed direction is to reap the benefits of ensemble modeling (explanation alignment, improved predictive performance, robustness against overfitting or dataset shift, etc.), while reducing the computational cost associated with operating large numbers of constituent models. Methods such as weight averaging (Izmailov et al., 2018) or model fusion (Singh and Jaggi, 2020) could potentially achieve this goal.

Furthermore, examining the effects of distillation and self-distillation could provide additional insights into model indeterminacy and the consistency of model explanations. Distillation techniques aim to compress the knowledge of an ensemble into a single, often simpler, model. Analyzing the impact of these techniques on the consistency and quality of explanations is an exciting prospect for future investigations. A connection to Bayesian methods might also be made, which inherently capture model uncertainty and can provide a measure of variability in single modes. However, these methods often face challenges related to computational efficiency and robustness to dataset shift. By contrast, ensemble methods can potentially offer a more widely accepted and practical alternative. The balance between ensemble techniques and Bayesian methods and their respective impacts on explanation consistency is worth considering.

**Broader impact**    The presented ensembling strategies for improving explanation consistency, while not explicitly designed to address issues of fairness or bias in AI models, have the potential for significant societal impact. It is expected that the insights gained from our methods will be beneficial for practitioners seeking to construct ML models that provide reliable explanations, which is especially crucial in fields where decision-making based on model outputs has high-stakes consequences, such as healthcare, finance, and criminal justice.

While the methods we showcase make strides in explanation consistency, they do not explicitly handle the problem of fairness and bias in AI. Model explanations that are consistent across different models might still be biased or unfair if the underlying models or data exhibit these issues. Thus, it's important to supplement our techniques with methods explicitly designed to test for and mitigate bias and unfairness in AI models.

Moreover, our approach does not guarantee perfect explanation consistency; there may be cases where explanations between models or different runs might still vary to a certain degree. This could have implications in scenarios where consistent explanations are particularly important, such as in the application of machine learning models in legal or healthcare contexts.

Given these broader impacts, it's crucial for practitioners applying our methods to be aware of these considerations. We recommend combining our ensembling techniques with existing and future fairness-aware methods and robust validation procedures to ensure comprehensive analysis of explanations. As this field evolves, we encourage future research on how to effectively integrate consideration of explanation consistency, fairness, and bias in AI model development and evaluation.

**Closing remarks**    In conclusion, our work provides a foundation to pave the way for future research directions towards leveraging neural network research to improve our knowledge of explanations amidst model indeterminacy. Although ensemble methods feature prominently in the neural network analysis literature, their application with respect to model explanations has received very little attention. As stated, advances in loss landscape understanding and implications of model indeterminacy have emerged as recent, yet distinct developments, with the two fields existing almost in parallel. We emphasize once more that the aim of this work is to serve as a catalyst for their convergence, and bring about a unified exploration of both areas. Through initial alignment of these directions, we have shed light on the interplay between model indeterminacy and the consistency of model explanations.

Moving forward, a rich and promising set of research directions awaits. Further exploration of the loss landscape, particularly in the pursuit of expedited mode connectivity, or distillation techniques, stands as an intriguing prospect. These directions would build on recent advancements in the field and potentially lead to more efficient ensemble construction. Additionally, the application of permutation symmetries to align model weights prior to averaging is another promising direction. Each of these techniques may not only improve computational efficiency but also enhance our understanding of the loss landscape and its relationship with model explanations. Such strategies are, in essence, leveraging the mathematical structure inherent in neural networks to expedite the exploration of the underspecification set and promote explanation consistency.

We believe that the pursuit of these directions will continue to push the frontiers of our understanding of neural networks, contributing positively to the development of reliable, trustworthy, and interpretable artificial intelligence systems.

