# OpenReview forum: "Consistent Explanations in the Face of Model Indeterminacy via Ensembling"
_ICML.cc/2023/Workshop/IMLH — IMLH 2023 Poster_

### Official Review · Reviewer_iuy6 · 2023-06-05

**Rating:** 7
**Confidence:** 3

**Review:**

In this paper a probe into the use of ensembling models based on weight perturbations is given. The paper is well written and has extensive experimentation to support the intuition of the probe. The limitations stated by the authors are true and if addressed we would gain significant insight into the explainability of ML models. Overall this is an interesting paper that I believe would start an interesting conversation during the workshop

---

### Official Review · Reviewer_C8eB · 2023-06-16
**This paper use ensemble methods to enhance the consistency of the explanations in model indeterminacy**

**Rating:** 8
**Confidence:** 3

**Review:**

This paper is well-organized. A combination of local and global ensembles can improve the performance compare to naive ensembling strategies. The experiment is convincing, and the conclusion is clear. Despite the limitations, I recommend acceptance.

---

### Official Review · Reviewer_Kp8r · 2023-06-19
**Well-written paper with complicated paper positioning**

**Rating:** 6
**Confidence:** 4

**Review:**

This paper addresses the issue of inconsistent explanations in the presence of model indeterminacy, where multiple equally performing models provide varying explanations for their predictions, and introduces ensemble methods to improve consistency. By exploring the underspecification set using insights from neural network loss landscapes, the proposed ensembling strategies effectively enhance explanation similarity. Experimental results on financial datasets demonstrate the efficacy of the ensembling methods in improving the consistency of explanations while maintaining computational efficiency.

The paper is well-written, and the experimentation is comprehensive. The introduction of mode connectivity and weight perturbation as practical improvements to the ensembling strategy is interesting. The results and discussion clearly support the efficacy of the proposed solution, and the paper is transparent and insightful about its limitations and future work.

However, a potential concern is the significance/existence of the problem itself. Multiple models performing similarly well but being "inherently" dissimilar may or may not be a major issue depending on the use case. The explanations—which is the focus of this paper—of such dissimilar models being dissimilar may be even less of an "inherent" issue (although from implication perspective this may be undesirable). The alignment of explanations may then be considered less faithful and consequently harmful. Therefore, the need and implications of 'forcefully' aligning explanations should be carefully considered.

---

### Meta-Review · Area_Chair_7R75 · 2023-06-19

**Recommendation:** Accept (Poster)
**Confidence:** 4

**Metareview:**

The paper investigates the XAI consistency problem and proposed novel ensemble methods of explanations with thorough experiments. The authors should consider addressing the reviewers comments, especially on the significance/existence of the problem and revise the motivation or limitation accordingly, for example, by emphasizing on uncertainty instead of enforcing consistency.

---

### Decision · Program_Chairs · 2023-06-20

Accept (Poster)